# Fabrication of Novel Heterostructure-Functionalized Graphene-Based TiO_2_-Sr-Hexaferrite Photocatalyst for Environmental Remediation

**DOI:** 10.3390/nano13010055

**Published:** 2022-12-22

**Authors:** Kefayat Ullah, Won-Chun Oh

**Affiliations:** 1Department of Applied Physical and Material Sciences, University of Swat, Khyber 19120, Pakhtunkhwa, Pakistan; 2Department of Advanced Materials Science & Engineering, Hanseo University, Seosan-si 31962, Chungnam, Republic of Korea

**Keywords:** photocatalysis, graphene, dyes, remediation, TEM

## Abstract

Novel visible-light photocatalyst (titanium-dioxide-functionalized graphene/strontium-hexaferrites) TiO_2_-FG/Sr-hexaferrite nanocomposites were fabricated using a simple hydrothermal technique. X-ray diffraction (XRD), field emission scanning electron microscopy (FESEM), diffuse reflectance spectroscopy (DRS), transmission electron microscopy (TEM), Raman spectroscopic analysis, and atomic force microscopy were used to analyze the composites as prepared. The unique TiO_2_-FG/Sr-hexaferrite-based composite catalyst reveals superior photocatalytic properties for the disintegration of organic dyes methylene blue (MB) and rhodamine B (Rh. B) under visible-light irradiation. The result showed that the functionalized graphene with ternary structure improved the catalytic behavior of the composite due to the synergistic effect of the TiO_2_-FG boosted by the graphene surface to provide a fast conducting path to the photogenerated charge carrier. The markedly high photocatalytic behavior has been ascribed to the formation of the ternary structure between TiO_2_, FG, and Sr-hexaferrites through interface interaction. The prepared photocatalyst composite exhibited better recyclability, which further confirms its future uses as a photocatalyst in industrial waste products.

## 1. Introduction

Environmental remediation through advanced nanomaterials attracts scientists globally. Heterostructure catalyst materials based on two semiconductors supported by a moderator have been widely used in the photocatalysis treatment process with many successful results [1]. The isolation of photoinduced electrons/holes exhibiting considerable oxidoreduction potentials on various semiconductors is a key benefit of heterogeneous systems [2]. A new donor/acceptor pair was therefore added to these systems as a redox mediator to help the photogenerated charge transportation between the semiconductor materials. These donor/acceptor mediators are mainly Fe^3^, Fe^2^, IO^3^, etc. [1,3,4,5]. Similarly, to minimize the backward reduction involving the redox mediator, a solid-state electron moderator was also devised [6,7]. Transition metal oxides (TMOs) are an innovative class of nanomaterials, often used as catalysts in various environmental photodegradation systems because of their typical properties, i.e., numerous active sites, minimum diffusion pathways, and low production and reaction mechanisms [8]. There are various disadvantages to using these metals/metal oxides as electron mediators, including their high cost, metallic pollution and corrosion, etc. In order to create a new heterogeneous system with increased photocatalytic effect for the degradation of environmental waste, a suitable electron mediator is therefore required. Carbon derivative graphene has been used widely to combine various semiconductor materials. The graphene in these composites acts as bridging between two semiconductors. The surface functionalization of graphene is a very important issue, and scientists need to handle it carefully [9]. The functionalities on the surface of graphene greatly affect the particle size of the guest material. The shape and size play an important role in the photocatalytic mechanism. Similarly, the modification of graphene’s surface through various semiconductor materials will pave the way for new semiconductor photocatalysts. These photocatalyst materials will have desirable properties. Graphene has already been reported as a photosensitizer in various semiconductor photocatalysts [10] and to have a great impact on the band gap of semiconductors, with increased absorption towards the visible part of the spectrum [11,12].

Integrated graphene/TiO_2_ complexes have drawn a lot of attention because of the large surface area as well as better charge conductivity of graphene and the nontoxic, affordable, and ecologically friendly behavior of TiO_2_ [13,14]. It has been demonstrated that graphene may effectively transfer and store photogenerated electrons when paired with a semiconductor like TiO_2_. It is due to the π-π interaction, which greatly enhances the affinity with organic pollutants to increase the efficiency of the photocatalyst. To meet the environmental requirements, special care must be taken when creating new photocatalysts, i.e., the easy and fast synthesis route and nontoxic behavior. Because the system of heterogeneous photocatalysts works together synergistically, they can create greater catalytic activity [15]. Quantum confinements allow the system energy level to be changed to the desired level. The exchange of electrons between two catalysts is the key driver of the photocatalytic reaction in a ternary system. The insertion of an electron mediator with strong conducting properties and significant interface contact poses the biggest challenge in such systems [13,16]. Despite these encouraging properties, the main problem in compound systems is the accumulation of nanostructures over graphene sheets because of the numerous functional groups found mostly on the surface of graphene. These agglomerations reduce the surface area and the photocatalytic efficiency of the material. Therefore, ternary photocatalyst materials must have a uniform distribution of nanoparticles, large surface area, and greater interfacial contact [17]. In this work, we report a facile route for the preparation of TiO_2_-FG/Sr-hexaferrite nanocomposites supported by hexaferrite nanoparticles. For the first step, graphene oxide was taken and functionalized using chemical vapor deposition techniques. The functionalized graphene (FG) was then used as an electron mediator in a ternary system to boost the catalytic behavior of the nanocomposites [18].

## 2. Experimental Section

### 2.1. Materials

Titanium (IV) n-butoxide (TNB, C_16_H_36_O_4_Ti) used as a titanium precursor was purchased from Samchun Pure Chemical Co., Ltd., Pyeongtaek-si, Korea. Strontium nitrate tetrahydrate H_8_N_2_O_10_Sr and ferric hydroxide Fe(OH)_2_ were purchased from Daejung Chemicals Co., Ltd., Siheung-si, Korea. The supplier of ethylene glycol was Dae-Jung Chemical and Metals Co., Ltd., Siheung-si, Korea. Dae-Jung Chemical and Metals Co., Ltd., Korea provided benzene, which was utilized as a precursor to carbon. Argon gas was purchased from Samchun Pure Chemical Co., Ltd. All chemicals were used directly without further purification, and distilled water was also used in every experiment. Graphene oxide was produced using the Hummers–Offeman method [19]. A specially designed tube furnace, consisting of an inner tube and outer tube furnace, were also used, as indicated in Figure 1. There are two nozzles, one for argon entry and another for benzene vapors. The inner furnace, with a temperature of 550 ^o^C, was used to functionalize graphene through benzene vapors. Benzene vapors were produced prior to heating at below its boiling temperature. The control valve was used for the proper amount of benzene inlet and argon in the furnace, as reported in our previous work [9]. The crucible containing GO was then immersed when the inner furnace temperature reached 550 ^o^C. The benzene flow was maintained for only 10 min, and the sample was then lowered to room temperature. The functionalized graphene was then used to prepare heterostructure composite materials. 

### 2.2. Preparation of TiO_2_-Functionalized Graphene

Titanium alkoxide precursor materials were used in the appropriate ratio with FG in ethylene glycol under hydrothermal conditions to obtain the composite. The solvent was sealed in an autoclave for around 3 h at 200 ^o^C. The obtained slurry type was then dried at 100 ^o^C to obtain TiO_2_/FG powder. The surface of the FG, as prepared, was treated with 60% HNO_3_ for 1 h. 

### 2.3. Fabrication of TiO_2_-FG/Sr-Hexaferrite Composites 

The ultrasonication techniques were used to synthesize TiO_2_-FG/Sr-hexaferrite nanocomposites. In accordance with the stoichiometric ratio, appropriate amounts of strontium nitrate tetrahydrate H_8_N_2_O_10_Sr and ferric hydroxide Fe(OH)_2_ were dissolved in EG under magnetic stirring for 2 h at 60 ^o^C to obtain the homogeneous solution. The obtained solution was then added to a Teflon-type autoclave and heated at 250 ^o^C for 12 h. The pH was adjusted to 7 by using a NaOH drop in the solution. The obtained powder was then calcined at 850 ^o^C. 

### 2.4. Characterization

X-ray diffraction (XRD) and field emission scanning electron microscopy (FESEM) were used to analyze the produced sample. Photocatalytic performance of all samples was evaluated for the methylene blue (MB) and rhodamine B (Rh. B) dyes under visible light irradiation. Raman investigation was carried out to look at the traces of graphene. A “labRam Aramis” Horiba Jobin Yvon spectrometer was used for the measurement. An argon-ion laser at 514 nm was utilized for the measurement. In order to conduct the morphological analysis, a field emission scanning electron microscope (FESEM (JSM)-5200 JOEL, Japan) with enhanced imaging capabilities which could be customized to performance needs was employed. High-resolution transmission electron microscopy (HRTEM JEOL, JEM-2012, Japan) was used to determine the state and the sheet morphology of the prepared graphene. The number of graphene layers and their stacking condition on the individual samples were assessed using a transmission electron microscope (TEM) at an acceleration voltage of 200 kV. The photocatalytic reactor was used, and a visible light bulb (150 W) with a filter (420 nm) was used as the light source. To determine the equilibrium between absorption and desorption, the sample was held in the dark. As a result, the relative efficiencies of each substance were assessed. A UV-visible spectrophotometer was used to measure the dye concentration (UV-2100, Shimadzu).

## 3. Results and Discussion

The results of the analysis using XRD techniques to identify the crystal phases are shown in Figure 1a,b. The XRD pattern of graphite, GO, and FG graphene is shown in Figure 1a. The TiO_2_ and strontium hexaferrites, as prepared, have a diffraction pattern that resembles the M-type hexagonal structure and anatase crystal phases, as shown in Figure 1b. The XRD peak of GO is shifted to (002), located at 26° at 2θ, confirming the formation of FG and supported by our previous reports [9]. The XRD peaks of Sr-hexaferrites (006) and FG (002) located at 26° at 2θ in Figure 1b, are difficult to differentiate because of the extensive reflection from the functionalized graphene surface. The Sr-hexaferrite composites exhibit the characteristic (110), (107), (210), (114), (203), (205), (206), (1011), (217), (2011), and (214) reflections that correspond to the crystal phase (JCPDS PDF#: 00-024-1207) [20]. The composite contains typical reflections, (101), (004), (200), (105), (211), and (220), which correspond to the anatase crystal phase of TiO_2_ referred to (JCPDS PDF#: 00-021-1272) [21], as shown in Figure 1b. The figure displays a slight reduction in intensity of the TiO_2_-FG/Sr-hexaferrite nanocomposites as compared to TiO_2_ and bare Sr-hexaferrites. On the surface of the FG sheet, we suppose that the suppression of peaks causes the crystalline phase of semiconductor materials to deform. The phonon confinement effect is produced by the interaction of nanoparticles with a graphene surface by reducing the formation of spherical-shape nanoparticles [22]. Therefore, as seen in TEM pictures of our ternary composites, we found some tube kind of Sr agglomerated and mixed-shape nanoparticles in the composite [23].

FSEM was used to study the morphological structure of our nanocomposites, as seen in Figure 2a–d. The SEM scan of bare GO is presented in Figure 2a. The image shows that the graphene sheets broke off in various directions in the absence of CVD treatment. This reveals that the formation of chain epoxy assemblies across graphene surfaces is responsible for the huge number of functional groups on the GO surface affecting the mechanical properties of graphene [24]. The SEM picture of the FG produced using CVD methods is shown in Figure 2b. This image demonstrates the successful functionalization of graphene using our methods. As illustrated in Figure 2b–d, the FG sheets were observed in arrangements of a few layers without breaking off in different directions [25]. According to earlier studies, functionalizing with organic solvents causes the aromatic chain to be perturbed, which is beneficial for electrical and optical properties.

The use of FESEM to examine the surface topography of our nanostructured materials is shown in Figure 3a–d. The complete morphological profile of FG in the composite was described using the SEM image; while the size and shape of nanoparticles are challenging to see using SEM, the TiO_2_-FG nanocomposites show the graphene’s plate-like formation loaded with TiO_2_. The overall images in Figure 3a–d make it evident that graphene is a sheet-like form that is split off in many directions [26]. The images in Figure 3b–d show TiO_2_-FG/Sr-hexaferrites at various magnifications in order to find the morphological information of the composite. After the functionalization of the graphene surface, retaining the sheet’s original morphology, the FG surfaces are covered with TiO_2_ and Sr-hexaferrite nanoparticles [27]. Functionalities on the graphene surface result in the Van der Waals interaction, which favors the accumulation of graphene sheets. Nanoparticle anchoring was useful in overcoming these interactions. The strontium and hexaferrite nanoparticles were found with minor accumulation because of the oxides and functional groups present on the surface of the graphene sheet. The bright spot refers to the TiO_2_ nanoparticles reported elsewhere. We assumed that the graphene surface that is host to the Sr-hexaferrite nanoparticles also includes TiO_2_. The SEM images of the sample after degradation are shown in Figure 3d,e. The morphology seems to be further broken off, maybe due to the adsorption of dye molecules and the decomposition after the visible light exposure. The morphology of these TiO_2_ and Sr-hexaferrite nanoparticles is supported by TEM images.

Figure 4a shows a TEM image of the TiO_2_-FG nanocomposites. The images shows that TiO_2_ nanoparticles are well distributed on the graphene sheet. Small-magnification TEM pictures were collected to reveal the extensive surface structure, distribution of nanoparticles, and graphene sheet composition. Figure 4b shows images of the Sr-hexaferrite nanocomposites at a 50 nm magnification. From these images, tube-type morphology can be seen with partial agglomeration. We presumptively attribute the apparent coagulation to the Sr nanoparticles’ varying size. To examine the size, shape, and distribution of nanoparticles, we performed the 200 nm and 500 nm resolution TEM images of TiO_2_-FG/Sr-hexaferrites, as shown in Figure 4b–d. As observed from these images, the TiO_2_ nanoparticles were found supporting Sr-hexaferrites on the graphene sheet. Thus, partial agglomeration of these images may be due to the ferrite’s attachment on FG sheets [28].

Figure 5 depicts the Raman bands of the FG-TiO_2_ nanocomposite, which exhibit a distinct G band at 1580 cm^−1^ and D band at around 1350 cm^−1^. The G and D bands in Figure 5 represent the vibration of a carbon atom at a disordered or defective site and the vibration of a carbon atom in plane, respectively, in our nanocomposites [29]. The D peak intensity is higher than that of TiO_2_-FG and TiO_2_-FG/Sr-hexaferrites, which further supports the functional density on the FG surface. There is a small 2D band observed in functionalized graphene which further confirms the layer-type morphology of FG sheets. The literature makes it abundantly evident that the D band results from the flaw caused by the functional groups on the carbon’s basal planes [30]. There are some signature peaks, such as Eg(1), B1, and g(1), found in the composite, which indicate anatase TiO_2_ vibrational modes [31]. The other metallic components will have a lower intensity and cannot be seen in Raman spectra. The intensity ratio of their D to G bands can be used to determine the order of defects in FG. These results demonstrate that the addition of ternary composites changes the D band intensity, while the values of the defects may vary, and a modest increase in the band intensity is observed. There are a number of potential causes for the shifting of the Raman bands of the composite with graphene materials, including the doping of the carbon with semiconductors, the laser excitation used during Raman experiments, etc. [32]. 

The DRS spectra of the composites are shown in Figure 6. An evaluation of the absorption spectra of TiO_2_-GO and TiO_2_-FG is given in Figure 6 to check the absorption edge of the composites. It is clear that the functionalization greatly affects the absorbance properties. There is an enhanced absorbance towards the visible region of the composites having FG. However, the GO-based TiO_2_ gives lower absorbance than our nanocomposites. The TiO_2_-GO sample was only included to check and compare the absorbance level of the two composites. This improved absorbance may be credited to the optimum loading and consistent distribution of nanoparticles on the graphene sheet. The sheet type morphology of graphene is better for providing a large number of reactions sites. Figure 6 shows that by including TiO_2_-FG in our graphene-based nanomaterials, its sharp edge in the visible range has been improved. The unpaired π electron in FG may interact with metal or semiconductor nanoparticles, shifting the band edge and causing more light to be absorbed in the visible spectrum [33]. The shift in absorption edge towards the visible portion of the electromagnetic spectrum by utilizing titanium-based carbon composites was also reported elsewhere [34].

We transformed the reflectance plot using the following equation: (hνα)^1/n^ = A(hν − E_g_),(1)
where h, ν, α, and E_g_ are the known constant, i.e., Planck constant, the frequency of vibration, the absorption coefficient, and the band gap, respectively, and A is a proportional constant. The quantity F(R∞) is related to the absorption coefficient and dedicated to the vertical axis of the spectrum. The α in the Tauc plot is replaced with the function F(R∞). The actual relational expression becomes
(hν F(R∞))^2^ = A(hν − E_g_)(2)

Using the KM function, the (hν F(R∞))^2^ was plotted against the hν. The specific band gap of our nanocomposite is defined by the curve with (hν F(R∞))^2^ and hν on the vertical axis [35]. The predicted band gap of the nanocomposites is defined by a line drawn tangent to the curve’s point of inflection, and the value of hv defines the band gap. The energy gap was found to be 2.091 eV, 2.3 eV, and 2.8 eV for TiO_2_-FG/Sr-hexaferrites, TiO_2_-FG, and TiO_2_-GO, respectively. The obtained plots are given in Figure 6.

The height and thickness of the TiO_2_-FG/Sr-hexaferrite composites was measured using AFM topography. Figure 7 depicts the AFM image of the TiO_2_-FG/Sr-hexaferrites. The sample provides the sheet-type graphene surface with some dot regions that are thought to be valleys between the graphene domains. The image illustrates some tube-type sharp edge on the graphene surface; this may be Sr nanorods, very useful for creating links with nanoparticles and their attachment. The TEM image has some minor evidence of variable shape in the Sr-hexaferrite composite. The surface architecture of the TiO_2_-FG/Sr-hexaferrites exhibited by AFM lead to the creation of nano heights, a distinctive and possibly complex phenomenon. The temperature inside the CVD, the reducing environment, the decomposition of oxides during the process, and benzene disintegration are just a few of the contending elements that could lead to the creation of such a structure [36].

### Photocatalytic Decolorization of Rh. B and MB Dyes

Two organic dyes, rhodamine B (Rh. B) and methylene blue (MB), were catalytically decomposed by the nanocomposite as prepared under the effect of visible light in order to ascertain the material’s catalytic performance. Three crucial processes are involved in the degradation and photocatalysis mechanisms; adsorption of pollutants onto catalyst materials, quick transfer of charges to produce radical species, and absorption by the photocatalyst are all necessary for the oxidation of organic pollutants. Due to its two-dimensionality and conjugated structure, FG serves as an adsorption and fast-transfer material in our nanocomposites [37,38]. The changes in concentration of the Rh. B and MB solution were observed in order to confirm the absorption ability our nanocomposites, as shown in Figure 8a–c.

To achieve adsorption and desorption equilibrium, the catalytic chamber was left in the dark for two hours. The chamber was then exposed to vis-light for 30 min after adsorption–desorption equilibrium was reached. The sample was collected for analysis, and the solids were collected using a centrifuge. A UV-vis spectrometer was used to analyze the sample activity. The photogenerated electrons reacted with the adsorbed oxygen to produce free radicals [39,40,41,42,43]. The photogenerated charge carrier reacted with the absorbed dye molecules, and subsequent mineralization occurred by producing CO_2_ and H_2_O as a result of the free radicals’ reaction with it. Figure 8c shows that nearly 80% of the dye molecule was degraded by the TiO_2_-FG/Sr-hexaferrite composite. The degradation can be seen as a gradual decline in peak intensity of the Rh. B dye at 554 nm and the MB dye at 595 nm [44,45]. The homogeneous dispersal of the TiO_2_ nanoparticles (by offering a high number of reaction sites) and the synergistic interaction between FG and Sr-hexaferrites as cocatalysts are responsible for the drop in concentration and better catalytic effect [46,47,48]. 

Cyclic tests were carried out to highlight the stability of the TiO_2_-FG/Sr-hexaferrite composite. Figure 9 shows that despite the fact that Rh. B and MB degradation occurred three times, the photocatalysts did not appear to lose any of their photocatalytic activity. This demonstrates that the TiO_2_-FG/Sr-hexaferrite photocatalyst has remarkable stability and was not photocorrodable throughout the catalytic oxidation of the Rh. B and MB molecules. Thus, it is possible that the composite photocatalyst composed of TiO_2_-FG and Sr-hexaferrites will be used in actual environmental purification procedures [49,50]. By examining its additional physical and chemical characteristics, our work suggests that the combination of TiO_2_-FG/Sr-hexaferrites can be utilized as an effective catalytic material [51].

## 4. Conclusions

In conclusion, ternary TiO_2_-FG/Sr-hexaferrites were synthesized using a simple and easy hydrothermal method. Successful CVD modifications were made to functionalized GO. The spherical TiO_2_ and Sr-hexaferrite dispersed well on the FG surface, as supported by TEM images, and were found beneficial for the degradation of organic dyes. The presence of minor 2D and some metallic peaks in the Raman spectra supports the existence of ternary TiO_2_-FG/Sr-hexaferrite nanocomposites. The functionalized graphene in the system serves as a stable heterogeneous photocatalyst material and electron mediator/absorber to retain the ternary system. The optical properties show red shift towards the visible region of the electromagnetic spectrum by their respective band gaps from DRS spectra. The strength and effectiveness of our recently created, ternary, graphene-based nanocomposites are demonstrated by the outstanding photocatalytic effect employed to degrade the organic dyes Rh. B and MB. The present work provides a new route for functionalizing graphene as a support material for the attachment and distribution of nanoparticles on the surface of two-dimensional sheets. In summary, in order to observe enhanced adsorption capacity of the nanocomposites, the FG-based composite may be the ideal material for future environmental remediation.

## Data Availability

The data are available upon request.

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
