# Peer review of "Fabrication of Novel Heterostructure-Functionalized Graphene-Based TiO2-Sr-Hexaferrite Photocatalyst for Environmental Remediation"

_nanomaterials, 2022, doi:10.3390/nano13010055_

Round 1

Reviewer 1 Report

The paper describes “Fabrication of novel heterostructure Functionalized graphene-based TiO2-Sr-hexaferrites photocatalyst for environmental remediation”. The work is the authors have covered exciting and reasonable effort. According to my consent, the quality of the manuscript is publishable in “MDPI-Nanomaterials.” Some flaws are needed to be improved before its publication. Below are my comments on this manuscript.

Comments:

1.         Ensure all abbreviations are written out in full the first time they are used. This is particularly important in the abstract and the conclusions, but it carefully works through the entire manuscript from this perspective.

2.         The author needs to re-plot the XRD in the correct form.

3.         The author needs to provide the EDX and elemental mapping.

4.         The author needs to provide the SEM after the degradation of dyes.

5.         Please rewrite the Conclusions. This section should include A summary of your key findings and your vision for future work.

6.         The typos and grammatical errors are scattered throughout the paper and must be corrected with the utmost care.

Author Response

The paper describes “Fabrication of novel heterostructure Functionalized graphene-based TiO2-Sr-hexaferrites photocatalyst for environmental remediation”. The work is the authors have covered exciting and reasonable effort. According to my consent, the quality of the manuscript is publishable in “MDPI-Nanomaterials.” Some flaws are needed to be improved before its publication. Below are my comments on this manuscript.

Question 1. Ensure all abbreviations are written out in full the first time they are used. This is particularly important in the abstract and the conclusions, but it carefully works through the entire manuscript from this perspective.

Response: Thank you for your comments. The abbreviation has been written and highlighted in the revised manuscript. 

Question 2. The author needs to re-plot the XRD in the correct form.

Response: The XRD has been replotted in sperate such that the data can be easily visible. As highlighted in figure 1(a-b).

Question 3. The author needs to provide the EDX and elemental mapping.

Response: Thank you for your comments, it will take some time to prepare more experimental procedure of the sample, at this stage it is very difficult for us to conduct more experiments, actually we are short of sample. Thank you so much for the comments.

Question 4. The author needs to provide the SEM after the degradation of dyes.

Response: The SEM images of the sample were provided according reviewer comments. As depicted and highlighted in figure 3.

Question 5. Please rewrite the Conclusions. This section should include A summary of your key findings and your vision for future work.

Response: The conclusion has been revised according to reviewer suggestion, with highlighted format in the original manuscript.

Question 6. The typos and grammatical errors are scattered throughout the paper and must be corrected with the utmost care.

Response: Thank you for your comments and careful investigation of our manuscript, many typos and grammatical mistakes were corrected.

Reviewer 2 Report

The authors describe a research article entitled “Fabrication of novel heterostructure Functionalized graphene-based TiO2-Sr-hexaferrites photocatalyst for environmental re-mediation”. The topic of the manuscript is interesting, and the manuscript constitutes an interesting article concerning the development of photocatalysts. A short conclusion highlighting the main results of this research article is also provided at the end of the document.

The work is well-written and a well-constructed introduction has been established by the authors. Sufficient spectra and figures are included in the manuscript for comprehension and clarity. Interesting and convincing results are also presented in this manuscript. Overall, I think that this is a manuscript that I recommend for publication after inclusion of minor revisions.

1) For the photocatalytic decolorization, what is the range of wavelength used ? please precise. Visible light is not sufficient. What is the light intensity ?

2) Authors focused on the mineralization of rhodamine B and methylene blue (MB). Numerous photocatalytic systems already exist for these two dyes.  Investigations of pollutants such as diclofenac, ibuprofen and other medicines would have been of higher interest for this study.

3) At present, no real evidence of formation of H2O and CO2 during mineralization is provided. The authors mentioned that around 80% of the dye are decomposed during irradiation (See Figure 8) What about the 20% remaining ? Especially, are the authors sure that no decomposition products more toxic than the initial product are formed. These 20% of remaining product should be carefully analysed.

Author Response

The authors describe a research article entitled “Fabrication of novel heterostructure Functionalized graphene-based TiO2-Sr-hexaferrites photocatalyst for environmental re-mediation”. The topic of the manuscript is interesting, and the manuscript constitutes an interesting article concerning the development of photocatalysts. A short conclusion highlighting the main results of this research article is also provided at the end of the document.

The work is well-written and a well-constructed introduction has been established by the authors. Sufficient spectra and figures are included in the manuscript for comprehension and clarity. Interesting and convincing results are also presented in this manuscript. Overall, I think that this is a manuscript that I recommend for publication after inclusion of minor revisions.

Question 1:  For the photocatalytic decolorization, what is the range of wavelength used? please precise. Visible light is not sufficient. What is the light intensity?

Response: The photocatalytic reactor was used, and a visible light bulb (150 W) with a filter (420 nm) was used as the light source. To determine the equilibrium between absorption and desorption, the sample was held in the dark. As a result, the relative efficiencies of each substance were assessed. A UV-visible spectrophotometer was used to measure the dye concentration (UV-2100, Shimadzu). The visible light photoctalysis is common to observe with graphene-based nanomaterials. The absorption towards the visible region provides sufficient evidences, that is why visible light is used in our current studies and reported elsewhere.  

https://doi.org/10.3390/catal7100305,

  1. Mater. Chem. C, 2020,8, 15940-15955,

 https://doi.org/10.1002/9781119641353.ch4

Question 2:  Authors focused on the mineralization of rhodamine B and methylene blue (MB). Numerous photocatalytic systems already exist for these two dyes.  Investigations of pollutants such as diclofenac, ibuprofen and other medicines would have been of higher interest for this study.

Response: Thank you for your comments, at the current stage we are short of prepared sample and cannot attempt more dyes or medicine to check the catalytic activity. We will consider the suggestions to our future work. Thank you again for valuable suggestions.

Question 3:  At present, no real evidence of formation of H2O and CO2 during mineralization is provided. The authors mentioned that around 80% of the dye are decomposed during irradiation (See Figure 8) What about the 20% remaining? Especially, are the authors sure that no decomposition products more toxic than the initial product are formed. These 20% of remaining product should be carefully analyzed.

Response: Thank you for your suggestions and comments, The sample was collected for analysis, and the solids were collected using centrifuge. UV-vis spectrometer was used to analyze the sample activity. The photogenerated electrons reacted with the adsorbed oxygen to produce free radicals[39-43]. The photo generated charge carrier reacts with the absorbed dye molecules and subsequent mineralization occurred by producing CO2 and H2O as a result of the free radicals' reaction with it. Figure 8(c) depicts that nearly 80% of the dye molecule was degraded by the TiO2-FG/Sr-hexaferrites composite. The degradation can be seen as a gradual decline in peak intensity of the Rh. B dye at 554 nm and the MB dye at 595nm [44, 45]. The percentage change may be due to the maximum absorption of dye molecules by our nanocomposites. which leads the maximum degradation of the organic dyes. The careful invetigation of the remaining dyes is a good point of concern dempending on various parametres. I hope we provide suffucient and suitable answeres to reviwer comments. And hope we will receive a postive repsonse.

Round 2

Reviewer 1 Report

Accept in the current form